# Plasma pesticide residues–serum 8-OHdG among farmers/non-farmers diagnosed with lymphoma, leukaemia and breast cancers: A case-control study

Arun Pandiyan[1], Summaiya Lari[1,2], Janardhan Vanka[1], Balakrishnan Senthil Kumar[1], Sudip Ghosh[1], Babban Jee[3], Padmaja R. Jonnalagadda[1] *

1 ICMR-National Institute of Nutrition, Tarnaka, Osmania University, Hyderabad, Telangana, India,
2 Department of Biochemistry, Acharya Nagarjuna University, Guntur, Andhra Pradesh, India, 3 Department of Health Research, Ministry of Health and Family Welfare, Government of India, New Delhi, India

* drpadmajaj@gmail.com

## Abstract

### Background

A hospital-based cross-sectional case-control study was conducted to investigate the association between exposure through various pesticide residues detected in the plasma and serum 8-OHdG levels among farmers and non-farmers diagnosed with leukaemia, lymphoma and breast cancers and compare the same with healthy controls with no cancer and no exposure.

### Methodology

The present study was conducted among the farmers and non-farmers visiting a regional tertiary cancer care hospital in Hyderabad, Telangana State, India. Data were collected by administering a pre-tested questionnaire through an interview followed by the collection of blood samples which were analyzed for pesticide residues using LC-MS/MS while the serum levels of 8-OHdG were measured using ELISA. Data were analyzed using SPSS 24.

### Results

The pesticide residues detected were chlorpyrifos, dimethoate, malathion, phosalone, and quinalphos which were approved and recommended for their use on the crops that were cultivated by the farmers in their plasma samples along with banned pesticide residues like monocrotophos, diazinon, and dichlorvos among farmers diagnosed with all three types of cancers while the non-farmers and healthy controls were not detected with any such residues. In addition, farmers diagnosed with leukemia had higher levels of all the pesticide residues in their plasma than those diagnosed with lymphoma and breast cancers. Further, a significant difference was also observed between profenofos residues in plasma and serum 8-OHdG levels.

**Data Availability Statement:** All relevant data are within the manuscript and its Supporting Information files.

**Funding:** This work was supported by Department of Health Research, Ministry of Health & Family Welfare, Government of India under grant R.11012/17/2017-HR The funders had no role in study design, data collection and analysis, decision to publish, or preparation of the manuscript.

**Competing interests:** The authors have declared that no competing interests exist.

## Conclusion

In the present study, though the farmers diagnosed with three types of cancers were detected with various types of pesticide residues analysed, only residues of profenofos showed a significant difference with serum levels of 8-OHdG suggesting the need for an in-depth follow up molecular studies in a larger cohort to assess the possible association between 8-OHdG levels with the pesticide residues among the exposed.

## Introduction

Agriculture is a crucial economic sector in India with more than 50% of the population depending on it for their livelihood [1]. The agricultural sector relies extensively on chemical pesticides for pest management and diseases during crop production. Occupational exposure to pesticides occurs during the preparation of agricultural fields through inhalation or dermal contact. Exposure to high concentrations over a short period of time can also lead to acute poisoning, but at times also have chronic effects, which may persist for several months to years after the initial exposure [2]. There are a number of occupational risk groups, ranging from individuals employed in the industrial production of pesticides and, in particular, the farm workers involved in mixing, loading, and spraying of pesticides on a regular basis as part of their farming activities. Farmers are reported to be continuously exposed to pesticides at levels significantly higher than the general population [3–6].

The current issue of hazard posed by pesticides to human health and the environment has raised concerns. The data for the last two decades regarding pesticide exposure and human health revealed that several pesticides cause neuronal disorders and degenerative diseases, some effect fetal growth and cause congenital anomalies while some can be carcinogenic to humans [7]. Over the past few decades, the extensive use and improper handling of pesticides in agriculture have caused serious threats to human health in many developing countries [8]. Human blood is the most accessible body fluid for ascertaining the pesticide residue levels though present even at very low concentrations while other sometimes remain accumulated to induce toxic effects while others may be metabolized after their entry into the human system and excreted [9]. However, the ultra-low quantities of residues left over un-metabolized in blood can induce adverse effects among exposed population. Therefore, the determination of pesticide residues in blood can be used as a biomarker for evaluating the ill effects of exposure. It has also been reported that persistent environmental contaminants such as pesticide residues have long been suspected to be implicated in cancer etiology [10]. Pesticides usually disturb the physiological and biochemical activities of lymphocytes and erythrocytes. Further, they can also act as endocrine disrupters; inhibit many enzymes, cause immune suppression and sometimes could even lead to cancer among susceptible individuals [11].

Of late, the increasing trend in the frequencies of various cancers has raised concerns in the scientific community to investigate on every possible cause of cancer, including the exposure to pesticides. Therefore, it is essential to investigate the possible association, if any, among those with occupational exposure to pesticides and potential cancer risks of these substances to humans. Pesticide-induced oxidative stress is known to have been associated with several diseases such as hormonal, neurodegenerative disorders, cardiovascular disease and cancer [12]. Oxidative damage to the genetic material under the influence of ROS generates a free radical induced oxidative lesion known as 8-hydoxy-2'-deoxyguanosine (8-OHdG) that play a vital role in the initiation and promotion of carcinogenesis [13]. 8-OHdG is known as the most

abundant DNA lesion as its not only relatively formed easily but also is promutagenic in nature, and hence considered as a potential biomarker for carcinogenesis [14]. Keeping in view of this, the present study was undertaken to assess the serum 8-OHdG and the pesticide residue levels in the plasma of the farmers exposed to pesticides prior to their diagnosis with leukemia, lymphoma and breast cancers and compare the same with non-farmers, who is having no exposure to pesticides but diagnosed with all three types of cancers and healthy controls who are neither having exposure nor diagnosed with above three types of cancers but belonging to family of farmers.

## Methodology

### Study design and ethical clearance

This hospital-based cross-sectional study was approved by the Institutional (NIN Protocol Number CR/08/II/2019) and hospital Ethics Committees (Reg. no.: ECR/227/Inst/AP/2013/RR-16). The data were collected from the farmers/non-farmers visiting a regional tertiary cancer care hospital in Hyderabad, Telangana State, India by administering the pre-tested questionnaire through an interview at the hospital premises during the periodical visits between 2018 and 2021. The participants were also explained the study objectives and written informed consent was also obtained from them before the collection of samples.

### Study population and sampling

Prior to the collection of data, a preliminary survey was conducted to assess the frequency and the types of different cancers reported in the regional tertiary cancer care hospital in Hyderabad, India. The findings from the preliminary survey revealed that the patients belonging to the farming community visiting the hospital were most commonly diagnosed either with leukaemia, lymphoma and breast cancer as compared to the other types of cancers which were recorded as relatively less. Based on the prevalence (81%) using the formula $n = (z)^2 p (1 - p) / d^2$, the estimated sample size calculated was 90 with a 5% level of significance at 10% precision. Stratified random sampling was used to assign the participants to a 2:1:1 ratio and this allocation supported the distinction between agricultural workers and other participants. Hence, the total number of participants was 360 i.e., 180 farm workers, 90 non-farm workers and 90 healthy controls were set as target for the present investigation.

### Inclusion and exclusion criteria

Farmers between the age group of 18 to 60 years with a previous history of involvement in farming activities and diagnosed with leukaemia (n = 60), lymphoma (n = 60), and breast cancers (n = 60) were included as cases, while non-farmers other than farmers (n = 90) diagnosed with similar types of cancers were selected as controls. The healthy controls (n = 90) were those with neither history of exposure to pesticides nor diagnosed with any types of cancers but belonging to the members of the farmer's families. Careful consideration was given while collecting the questionnaire data that none of the family members who had history of occupational exposure to pesticides were included under the healthy control groups. Subjects diagnosed with any other cancers were not considered for the study.

### Questionnaire data

A validated questionnaire prepared based on our previously conducted epidemiological studies was used to obtain information on different variables. It was divided into five sections with a total of fifty-seven questions. The first section included questions on socio-demographic

particulars such as gender, age and the extent of land holding by the farm workers, their principal occupation followed by their educational status while the second section constituted questions on the particulars of pesticide exposure such as spraying activities and duration of exposure followed by the third section consisting a set of questions on their involvement in other allied farming activities such as watering, sowing, cutting, thrashing, weeding followed by harvesting and the types, quantity and names of pesticides used by them in different forms for the cultivation of different crops such as wheat, paddy, cotton, and vegetables while the fourth section dealt with personal information such as dietary habits, consumption of alcohol and smoking and the respective quantities consumed per day. Further, the fifth section contained questions related to the participant's attitude, awareness and practice, knowledge on the routes of pesticides exposure, frequency of spraying and protective measures adopted such as use of PPE, if any, while handling pesticides, reading the precautions on the label, and storage/sanitary practices adopted such as washing hands and clothes immediately after handling the pesticides or reuse the same without washing etc.

## Pesticide residue analysis in plasma and tissue samples using LC-MS/MS

### Materials & chemicals

The certified reference materials of all the thirty pesticide standards viz., oxamyl, monocrotophos, imidacloprid, fenitrothion, dimethoate, propoxur, malaoxon, dichlorvos, carbofuran, carbendazim, carbaryl, imidan, methiocarb, malathion, phenthoate, chlorfenvinfos, pirimiphos methyl parathion, ethion, chlorpyrifos, carbosulfan, quinalphos, phosalone, chlormefos, diazinon, profenofos, acephate, fenubocarb, iprovalicarb, methomyl, anilophos, triethyl phosphate (Internal Standard) (all chemicals having >97% purity) and Nonidet P-40 were procured from M/s. Sigma Aldrich (St Louis, MO, USA) while the LC-MS grade solvents viz., methanol, acetonitrile, and formic acid were also obtained from M/s. Biosolve (Dieuze, France).

### Preparation of standard solutions

Pesticide stock solutions and internal standard (triethyl phosphate) were prepared in 50% acetonitrile (1 mg/mL) and stored at −20˚C and diluted serially to the required concentrations.

### Extraction procedure

The LC-MS/MS system used for the purpose of analysis was a Thermo TSQ Altis system coupled with Thermo 3000 RSLC with a quaternary gradient pump and an auto-sampler (Thermo Fisher Scientific, USA) (Box 1). The separation of pesticides was carried out using Agilent ZORBAX SB-C18 column of length 250 mm, 4.6 mm internal diameter and 5 μm of particle thickness. Water containing 0.1% formic acid and methanol with 0.1% formic acid were used as mobile phase A and B respectively in gradient mode with the initial composition of 10% B ramped at a uniform rate to 100% till 20 min and maintained at the same composition till 25 min and brought to initial conditions at 25.10 min and the same was maintained till 30 min. The flow rate of the mobile phase used was 800 μL/min throughout the gradient and the ionisation was carried out by APCI in positive ion mode with source temperature of 400˚C, 6 Volt of discharge current, 10 units of Auxiliary gas and 2 units of sheath gas flow rate. Further, the transfer line temperature was maintained at 375˚C and the mass spectrometer was calibrated once in five days and tuned to the utmost possible sensitivity using 1 μg/mL solution of triethyl phosphate before proceeding for the analysis. The resolution of equipment was maintained at

---

## Box 1. LC-MS/MS conditions

LC-MC/MS–Conditions

**Instrument:** Orbitrap LC-MS (Make: Thermo Fischer)

**Mobile Phase: A**–Water + 0.1% Formic Acid

**B**–Methanol + 0.1% Formic Acid

**Column:** Agilent ZORBAX SB (C18)–[250mm x 4.6mm x 5µm]

**Source:** APCI, Positive mode

**Solvent:** 50% Acetonitrile

---

50,000 FWHM throughout the analysis. The samples were then analysed in full scan mode in the mass range of 50–500 units. Plasma samples from human volunteers were obtained for the purpose of standardization. The stock solutions were serially diluted to primary standards in the range of 20, 10, 5, 2, 1, 0.5, 0.2, 0.1, 0.05, 0.02 and 0.01 µg/mL concentrations using 50% acetonitrile (in water) and were stored in –20˚ for their subsequent use. About 200 µL of human plasma samples were spiked with the primary standard solutions of certified reference materials (CRMs) at 1, 2, 5, 10, 20, 50, 100, 200, 500, 1000 and 2000 ng/mL concentrations. Triethyl phosphate (TEP) at the concentration of 200 ng/mL was added as an internal standard to the above CRMs. The simple liquid-liquid extraction process was followed for the extraction of samples and the procedure is as follows: the spiked plasma samples were mixed with 200 µl of acetonitrile and were vortexed for 5 minutes. The supernatant solution was carefully transferred to another 2 ml vial and the solution was completely evaporated to dryness using a speed vacuum concentrator for 12 hours (overnight) (M/s. Eppendorf, Germany). The residues were reconstituted using 200 µl acetonitrile and then analysed using LC-MS/MS (Table 1).

## Method validation

The analytical method validation was performed according to the ICH Q2 (R1) guidelines. The limit of detection (LOD) of the method for each pesticide residue was measured at signal to noise ratio S/N of $\geq$ 3:1 while the limit of quantification (LOQ) was measured at signal to noise ratio S/N of $\geq$ 10:1 and the linearity of the method was tested over the concentration range of 1 to 2000 ng/mL by spiking working standard solutions along with a known concentrations of internal standard (IS) at 200 ng/mL into the extracts of blank human plasma samples. The calibration curves were plotted for the ratio of the peak areas (Analyte/IS) against the concentration of the analytes. The results obtained showed satisfactory linearity with regression coefficients ($r^2$) greater than 0.99 for all the pesticide residues studied. Further, the intraday accuracies were determined by preparing the quality control (QC) samples at 10, 50 and 500 ng/mL concentration levels in six replicates (n = 6) and analysed along with the calibration samples within a day while the inter-day accuracies were also determined using quality control (QC) samples prepared afresh each and every day at three levels viz., 10, 50 and 500 ng/mL concentrations in six replicates (n = 6) and analysed along with the samples meant for calibration purpose for three consecutive days. The inter-day and intraday accuracies were observed to be in the acceptable ranges and the precision values evaluated in terms of relative standard deviations (RSD, %) observed to be less than 15% as per the ICH guidelines.

**Table 1. Optimized LC-APCI-MS/MS parameters for targeted pesticides.**

| S.No. | Pesticides | $R_t^{a)}$ (min) | M.W $^{b)}$ (g/mol) | $(M+H)^+$ | LC-QqQ-MS/MS | |
|---|---|---|---|---|---|---|
| | | | | | Quantifier | Qualifier |
| 1. | Fenobucarb | 3.67 | 207.27 | 208.13 | 95.00 | 152.05 |
| 2. | Acephate | 8.60 | 183.2 | 184.00 | 142.99 | 124.92 |
| 3. | Methomyl | 9.92 | 162.21 | 163.05 | 106.04 | 87.97 |
| 4. | Monocrotophos | 9.99 | 223.16 | 224.00 | 175.05 | 209.07 |
| 5. | Dimethoate | 11.02 | 229.26 | 230.00 | 124.91 | 198.97 |
| 6. | Malaoxon | 12.00 | 314.29 | 315.00 | 98.97 | 126.91 |
| 7. | Dichlorvos | 12.15 | 220.98 | 221.00 | 108.97 | 144.91 |
| 8. | Carbofuran | 12.22 | 221.25 | 222.00 | 122.98 | 125.05 |
| 9. | Carbaryl | 12.27 | 201.22 | 202.00 | 145.07 | 126.98 |
| 10. | Oxamyl | 12.27 | 219.26 | 220.00 | 146.04 | 163.05 |
| 11. | Carbendazim | 12.87 | 191.18 | 192.00 | 160.07 | 131.98 |
| 12. | Imidan | 12.90 | 317.31 | 318.00 | 160.07 | 132.98 |
| 13. | Malathion | 13.15 | 330.36 | 331.04 | 126.91 | 285.00 |
| 14. | Methiocarb | 13.18 | 225.31 | 226.00 | 121.00 | 169.05 |
| 15. | Fenitrothion | 13.25 | 277.23 | 278.00 | 246.00 | 218.00 |
| 16. | Iprovalicarb | 13.45 | 320.4 | 321.00 | 119.07 | 203.50 |
| 17. | Phenthoate | 13.63 | 320.35 | 321.03 | 119.07 | 203.15 |
| 18. | Anilofos | 13.69 | 367.9 | 368.00 | 199.47 | 125.48 |
| 19. | Chlorfenvinphos | 13.81 | 359.6 | 361.00 | 155.07 | 206.92 |
| 20. | Quinalphos | 13.92 | 298.29 | 299.06 | 147.07 | 163.00 |
| 21. | Diazinon | 13.92 | 304.35 | 305.10 | 169.05 | 153.05 |
| 22. | Pirimiphos methyl | 13.98 | 305.33 | 306.00 | 108.00 | 164.08 |
| 23. | Ethion | 14.42 | 384.48 | 385.00 | 198.97 | 142.88 |
| 24. | Profenofos | 14.54 | 373.63 | 374.94 | 302.90 | 344.88 |
| 25. | Chlorpyrifos | 14.83 | 350.59 | 351.93 | 197.95 | 333.07 |
| 26. | Imidacloprid | 14.89 | 255.66 | 256.06 | 175.09 | 209.05 |
| 27. | Carbosulfan | 16.04 | 380.5 | 381.00 | 117.97 | 160.15 |
| 28. | Propoxur | 19.42 | 209.24 | 210.11 | 168.05 | 111.04 |
| 29. | Phosalone | 20.82 | 367.81 | 368.00 | 119.89 | 170.98 |
| 30. | Chlormefos | 21.80 | 234.71 | 234.97 | 114.96 | 142.93 |
| IS | Triethyl Phosphate | 8.54 | 182.15 | 183.07 | 142.98 | 124.91 |

a) M.F–Molecular Formula, b) M.W–Molecular Weight, c) MRM–Multiple Reaction Monitoring, d) Base Peak.

To estimate the matrix interference and percentage reduction in signal, 200 μL of plasma were extracted as per the above-mentioned procedures, and the standard solutions of pesticides at 10, 50, and 500 ng/mL concentrations were added to the extracts and reconstituted using 200 μL of acetonitrile and analysed by LC-MS/MS. The data obtained were compared with the similar concentrations of the standard solutions prepared using acetonitrile solvent and it was found that the percentage ratio of signal to noise calculated had decreased and the data obtained indicated 20±8.5% reduction in the spiked matrix samples showing no effect of matrix interference. The matrix effect and recoveries were estimated using the equations given below:

Matrix Effect (%) = $(C_a/C_s)$ x 100

Recovery (%) = $(C_b/C_a)$ x 100

where $C_a$ = peak area of the analyte of the sample spiked with the target pesticides after extraction; $C_s$ = peak area of the analyte of standard solution; $C_b$ = peak area of the analyte of the sample spiked with the target pesticides before extraction.

The data on the percentage recovery of the analytes showed acceptable recoveries as per the pharmacopeia guidelines (80–120%) and further the matrix effect was also found to be satisfactory. In addition, the carryover studies were also performed to observe if there was any interference through pesticide standards or any other carryover in the blank runs plasma (1000 ng/mL) samples using a test sample spiked with highest concentration of six pesticides. The observed results showed that there was no carryover from the previous samples in the blank run.

### Determination of serum 8-hydroxy-2'-deoxygunosine (8-OHdG)

The serum levels of 8-OHdG were analyzed using commercially available ELISA kits (Elabscience, USA) as per the manufacturer's instructions provided in the leaflet. Briefly, 50 μL each of standard and serum samples were added in respective wells of a 96-well plate. Subsequently, biotinylated detection Ab (1:100 dilutions) working solution was added to the wells, followed by incubation for 45 min at 37˚C. After incubation, the plates were washed and aspirated thrice using a washing buffer to which 100 μL of HRP-streptavidin conjugate solution was added and were incubated again for 30 min at 37˚C. Subsequently, the plates were aspirated and washed again using washing buffer followed by the addition of 90 μL substrate reagent and incubated at 37˚C for 15 min. The incubated wells were added with 50μL of stop solution and the absorbance was read at 450 nm using a microplate reader (BioTek Instruments). The samples were estimated in replicates as per the standard protocol.

### Statistical analysis

Data were verified for completeness and consistency and analyzed using SPSS v28.0 (IBM SPSS Statistics for Windows, Version 28.0. Armonk, NY: IBM Corp.). The Shapiro-wilk test was performed to assess the normality of the data prior to conducting the analysis. Student t-test was performed to compare the mean levels of serum 8-OHdG while the violin plot to depict the mean levels of serum 8-OHdG between the farmers, non-farmers and the healthy controls was generated using Stata (Statacorp LLC, US). Student t-test was used to compare the levels of detected pesticide residues in plasma and serum 8-OHdG to assess the significant difference, if any. In addition, Pearson correlation was used to determine associations, if any, between pesticide residue levels detected in plasma samples and serum 8-OHdG. Association between serum 8-OHdG levels and other confounding factors such as age, gender, personal habits and duration of exposure to pesticides was determined using regression analysis. The data were considered significant when $p < 0.05$ at 95% CI.

### Results

Majority of the farmers (n = 76) were involved in wheat cultivation, followed by rice (n = 50). The remaining farmers were engaged in the cultivation of commercial crops like cotton (n = 35), vegetable crops (n = 11) such as brinjal and chilly and fruits (n = 8) like mangoes and guava. Apart from spraying pesticides, all the farmers reported to have been engaged in other allied agricultural activities viz., ploughing, cutting, watering, thrashing and weeding also. It was also found that more than 60% of the farm women diagnosed with breast cancer were also engaged predominantly in other agricultural activities too prior to their diagnosis. From the self-reported information collected on the usage of pesticides and the crops cultivated by them, it was found that four pesticides belonging to the class of insecticides viz.,

**Table 2. Demographic particulars of farmers diagnose with three types of cancers.**

| Variable | Leukaemia (n = 60) | Lymphoma (n = 60) | Breast Cancer (n = 60) |
|---|---|---|---|
| **Mean age (years)** | 38.92 | 44.07 | 45.45 |
| **Gender** | Male (n = 38)<br>Female (n = 22) | Male (n = 32)<br>Female (n = 28) | Male (n = 0)<br>Female (n = 60) |
| **Type of house** | Pucca (n = 20)<br>Semi pucca (n = 34)<br>*Kutcha* (n = 6) | Pucca (n = 10)<br>Semi pucca (n = 43)<br>*Kutcha* (n = 7) | Pucca (n = 19)<br>Semi pucca (n = 30)<br>*Kutcha* (n = 11) |
| **Location of the house** | In the farm (n = 6)<br>Away from the farm (n = 54) | In the farm (n = 8)<br>Away from the farm (n = 52) | In the farm (n = 14)<br>Away from the farm (n = 46) |
| **Educational status** | Illiterate (n = 18)<br>Read & Write (n = 41)<br>Primary (n = 1) | Illiterate (n = 19)<br>Read & Write (n = 31)<br>Primary (n = 10) | Illiterate (n = 26)<br>Read & Write (n = 34)<br>Primary (n = 0) |
| **Major occupation** | Own cultivation (n = 29)<br>Tenant cultivation (n = 8)<br>Agricultural labour (n = 20)<br>Other labour (n = 3) | Own cultivation (n = 12)<br>Tenant cultivation (n = 31)<br>Agricultural labour (n = 17) | Own cultivation (n = 4)<br>Tenant cultivation (n = 14)<br>Agricultural labour (n = 42) |
| **Family engaged in agricultural activities** | Yes (n = 19)<br>No (n = 41) | Yes (n = 36)<br>No (n = 24) | Yes (n = 31)<br>No (n = 29) |
| **Land holding in acre (mean)** | 0.958 | 0.533 | 0.117 |

monocrotophos, dichlorovos, chlorpyrifos and acephate were used by majority of the farmers (n = 54). Quinalphos was predominantly used for crops such as cotton and wheat while lambda cyhalothrin was used on tomatoes and brinjal. As per the self-reported information provided by them, for rice, the most commonly used pesticide for paddy was found to be chlorpyrifos. None of the farmers reported to have used any personal protective equipment such as face mask, gloves, apron and boots while during spraying. With respect to personal habits, all the farmers (n = 180) were found to be non-vegetarians, with an average consumption for once a week. None of the farm women diagnosed with breast cancer (n = 60) reported to have had the habit of smoking cigarettes, cigar or *beedies* followed by alcohol consumption. Among the farmers diagnosed with lymphoma, 15% of them reported to have consumed an average 90 ml of alcohol every day while 28% of the farm women diagnosed with breast cancer reported to have had the habit of chewing tobacco every day (Table 2).

## Pesticide residues in plasma samples

The chromatogram of different pesticides obtained from the LC-MS/MS analysis of the spiked plasma sample is presented in Figs 1 and 2. The standard curve obtained with concentration range of 1 to 2000 ng/mL was found to be linear with $r^2$ greater than 0.99 for all the thirty pesticides analyzed. The sensitivity of the method was evaluated by determining the LOD and LOQ and it was found that the LOD of the tested pesticides was ranging from 1 to 5 ng/mL and LOQ was between the range of 2 and 10 (Table 3). The results of inter and intra-days accuracies and precisions at three different concentration levels of pesticides were observed to be in the acceptable ranges and the precision values evaluated in terms of relative standard deviation (%RSD) obtained was less than 15% as per the ICH guidelines (Table 4). The variation coefficients of intra and inter-days ranged from 2.25 to 14.85%. The average recoveries were ranging between 85.56 and 120.25 for the selected pesticides in the present study.

A total of 360 blood samples were analysed for pesticide residues. Among the farmers diagnosed with leukaemia (n = 60), about 52% of the samples (n = 31) were detected with eleven organophosphates (0.18–1915.34 ng/mL), two carbamate (0.24–518.82 ng/mL) and one neonicotinoid (0.1–86.3 ng/mL) residues while among the 60 farmers diagnosed with lymphoma,

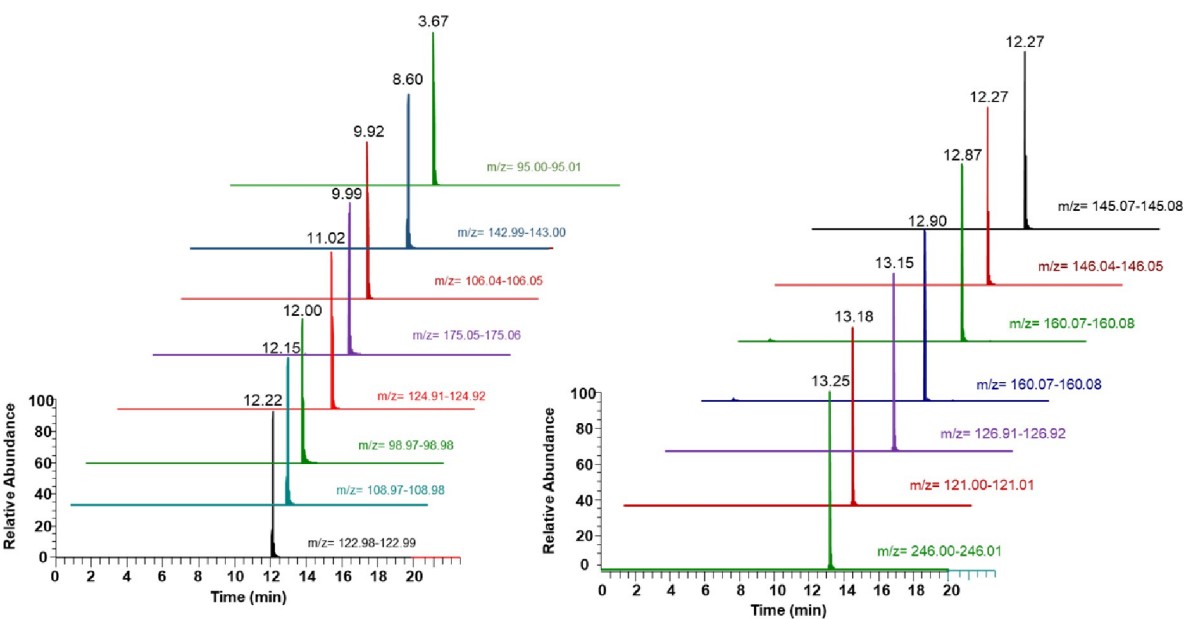

**Fig 1. Extracted ion chromatograms (XIC) of pesticides analyzed using LC-MS/MS.**

only 12 were detected with two organophosphate residues (0.64–72.55 ng/mL). Among the farm women (n = 60) diagnosed with breast cancers, only 2 were detected with organophosphate (2.11–17.64 ng/mL) and carbamate (17.12–23.6 ng/mL) residues each. None of the non-farmers and healthy controls was detected with any pesticide residues in their plasma samples (Table 5). When the farmers were categorized into two groups based on the number of years of exposure (i.e., high and low), it was observed that the data analyzed related to the levels of

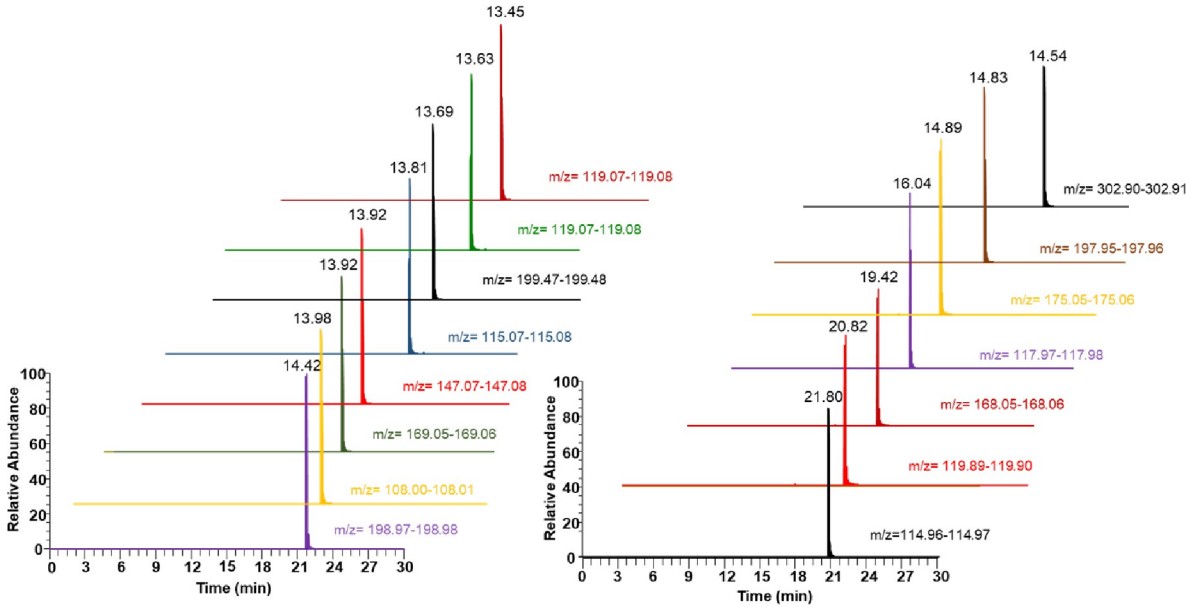

**Fig 2. Extracted ion chromatograms (XIC) of pesticides analyzed using LC-MS/MS.**

**Table 3. LC-MS/MS method validation parameters of pesticides obtained from the plasma samples.**

| S.No. | Pesticides | Linearity range (ng/mL) | LOD [b)] (ng/mL) | LOQ [c)] (ng/mL) | Validation accuracy (%RSD [d)]) | | |
|---|---|---|---|---|---|---|---|
| | | | | | 10 ng/mL | 50 ng/mL | 500 ng/mL |
| 1. | Oxamyl | 1–2000 | 2 | 5 | 98.75 (6.28) | 99.84 (11.01) | 101.24 (13.55) |
| 2. | Monocrotophos | 1–2000 | 2 | 5 | 100.02 (10.22) | 112.11 (9.65) | 102.45 (11.20) |
| 3. | Imidacloprid | 1–2000 | 1 | 2 | 97.85 (8.52) | 89.55 (5.66) | 111.24 (9.45) |
| 4. | Fenitrothion | 1–2000 | 2 | 5 | 98.66 (10.25) | 99.69 (12.53) | 97.25 (4.56) |
| 5. | Dimethoate | 1–2000 | 1 | 2 | 102.01 (3.56) | 101.11 (7.55) | 111.35 (2.25) |
| 6. | Propoxur | 1–2000 | 5 | 10 | 109.60 (12.36) | 104.44 (14.22) | 102.25 (13.08) |
| 7. | Malaoxon | 1–2000 | 1 | 2 | 100.23 (9.52) | 97.65 (9.56) | 99.63 (12.39) |
| 8. | Dichlorvos | 1–2000 | 1 | 2 | 87.58 (4.99) | 98.95 (7.52) | 96.54 (6.59) |
| 9. | Carbofuran | 1–2000 | 1 | 2 | 96.56 (3.74) | 103.54 (6.99) | 104.44 (5.88) |
| 10. | Carbendazim | 1–2000 | 2 | 5 | 98.96 (9.56) | 98.01 (12.58) | 96.12 (13.66) |
| 11. | Carbaryl | 1–2000 | 1 | 2 | 101.21 (14.8) | 100.05 (9.63) | 105.65 (4.59) |
| 12. | Imidan | 1–2000 | 1 | 2 | 85.56 (9.66) | 96.54 (7.55) | 98.99 (14.08) |
| 13. | Methiocarb | 1–2000 | 1 | 2 | 100.28 (5.66) | 102.22 (2.25) | 100.05 (11.69) |
| 14. | Malathion | 1–2000 | 1 | 5 | 98.60 (3.58) | 99.65 (9.64) | 100.24 (10.30) |
| 15. | Phenthoate | 1–2000 | 1 | 2 | 102.55 (9.44) | 100.45 (5.22) | 101.63 (4.08) |
| 16. | Anilofos | 1–2000 | 5 | 10 | 106.65 (14.20) | 111.05 (12.36) | 115.25 (14.23) |
| 17. | Pirimiphos methyl | 1–2000 | 1 | 2 | 116.52 (5.23) | 111.25 (9.65) | 100.45 (11.54) |
| 18. | Ethion | 1–2000 | 2 | 10 | 100.25 (3.25) | 120.01 (8.56) | 105.56 (7.83) |
| 19. | Chlorpyrifos | 1–2000 | 2 | 5 | 102.25 (8.55) | 100.23 (2.35) | 100.28 (14.59) |
| 20. | Carbosulfan | 1–2000 | 1 | 2 | 100.24 (9.55) | 101.28 (14.85) | 98.45 (5.68) |
| 21. | Quinalphos | 1–2000 | 1 | 2 | 98.63 (5.55) | 97.55 (13.59) | 91.25 (8.56) |
| 22. | Phosalone | 1–2000 | 2 | 5 | 105.64 (4.85) | 100.09 (12.58) | 120.25 (9.90) |
| 23. | Chlormefos | 1–2000 | 1 | 2 | 102.28 (13.56) | 100.56 (8.56) | 100.58 (12.54) |
| 24. | Diazinon | 1–2000 | 2 | 5 | 96.55 (12.85) | 96.32 (9.57) | 97.45 (11.08) |
| 25. | Profenofos | 1–2000 | 5 | 10 | 98.21 (10.02) | 91.56 (4.58) | 90.25 (4.59) |
| 26. | Acephate | 1–2000 | 2 | 5 | 110.25 (7.59) | 105.56 (9.32) | 117.85 (3.58) |
| 27. | Fenobucarb | 1–2000 | 2 | 5 | 111.25 (6.25) | 105.56 (11.02) | 118.96 (14.03) |
| 28. | Iprovalicarb | 1–2000 | 2 | 5 | 117.58 (9.55) | 114.25 (13.53) | 95.56 (12.52) |
| 29. | Methomyl | 1–2000 | 1 | 2 | 102.25 (10.25) | 111.25 (12.09) | 105.56 (5.59) |
| 30. | Chlorfenvinphos | 1–2000 | 1 | 2 | 105.56 (4.56) | 98.65 (6.52) | 99.54 (8.87) |

a) Rt–Retention time, b) LOD–Limit of Detection, c) LOQ–Limit of Quantification, d) The values in the parentheses represent %RSD valves (n = 6).

pesticide residues in the plasma samples were found to be relatively low among those having high exposure as compared to those having low exposure (Table 6).

## Serum 8-hydroxy-2-'deoxyguanosine

The blood samples collected from 360 subjects (180 farmers; 90 non-farmers; 90 healthy controls) when analysed for the estimation of serum 8-OHdG showed higher levels among farmers diagnosed with three types of cancers as compared to non-farmers diagnosed with similar types of cancers and healthy controls with no cancers and no exposure (Fig 3). There found a significant difference in the levels of serum 8-OHdG between farmers, non-farmers and healthy controls (p<0.01).

However, there was no significant difference observed in their levels between the high and low exposure groups among the farmers diagnosed with all three types of cancers (Table 6). Further, the regression analysis also showed no significant association between the duration of

**Table 4. The intra-day and inter-day mean percentage accuracies.**

| S.No. | Pesticides | Intra-day accuracy (SD±%RSD) | | | Inter-day accuracy (SD±%RSD) | | |
|---|---|---|---|---|---|---|---|
| | | 10 ng/g | 50 ng/g | 500 ng/g | 10 ng/g | 50 ng/g | 500 ng/g |
| 1. | Oxamyl | 0.4 ± 3.6 | 1.10±2.39 | 12.21±2.15 | 0.50±3.96 | 1.44±3.09 | 13.38±2.42 |
| 2. | Monocrotophos | 0.7 ± 4.3 | 4.59±8.56 | 23.35±4.16 | 0.39±3.32 | 1.30±2.28 | 14.66±2.37 |
| 3. | Imidacloprid | 0.4 ± 4.2 | 1.67±2.82 | 12.30±2.17 | 0.40±3.81 | 1.23±2.60 | 15.42±2.71 |
| 4. | Fenitrothion | 0.4 ± 3.5 | 1.62±3.79 | 27.91±4.89 | 0.42±3.00 | 2.22±4.81 | 26.69±4.34 |
| 5. | Dimethoate | 0.6 ± 4.8 | 4.25±11.21 | 19.59±3.88 | 0.43±3.65 | 1.80±3.25 | 17.56±3.06 |
| 6. | Propoxur | 0.4 ± 4.8 | 1.42±3.37 | 18.78±4.37 | 0.34±3.48 | 1.48±3.67 | 9.04±2.20 |
| 7. | Malaoxon | 0.4 ± 3.6 | 1.37±2.83 | 21.95±4.23 | 0.39±3.17 | 2.12±4.22 | 14.24±2.48 |
| 8. | Dichlorvos | 0.2 ± 5.2 | 2.70±4.99 | 20.31±3.08 | 0.87±3.36 | 1.68±4.23 | 22.15±3.04 |
| 9. | Carbofuran | 0.4 ± 3.4 | 1.47±2.71 | 21.03±3.96 | 0.49±4.01 | 1.63±3.16 | 14.20±3.64 |
| 10. | Carbendazim | 0.3 ± 4.6 | 2.26±4.21 | 26.47±5.49 | 0.18±2.26 | 1.22±3.53 | 18.16±3.15 |
| 11. | Carbaryl | 0.6 ± 4.8 | 1.95±4.78 | 18.31±3.35 | 0.61±4.62 | 1.81±4.45 | 19.43±3.20 |
| 12. | Imidan | 0.2 ± 2.9 | 2.79±4.64 | 15.68±2.89 | 0.27±3.30 | 1.94±3.22 | 14.87±2.60 |
| 13. | Methiocarb | 0.5 ± 4.5 | 1.65±3.53 | 25.55±4.63 | 0.59±4.79 | 1.90±4.11 | 22.43±4.16 |
| 14. | Malathion | 0.2 ± 2.2 | 2.51±4.26 | 12.39±2.28 | 0.56±4.93 | 2.84±4.06 | 14.63±2.70 |
| 15. | Phenthoate | 0.2 ± 3.0 | 7.79±13.40 | 26.64±4.96 | 0.41±4.43 | 2.58±3.65 | 23.44±4.58 |
| 16. | Anilofos | 0.4 ± 4.5 | 2.23±4.42 | 0.48±4.30 | 0.47±4.22 | 1.31±3.56 | 0.35±4.40 |
| 17. | Pirimiphos methyl | 0.7 ± 4.7 | 5.64±5.24 | 12.54±6.35 | 0.45±2.39 | 1.24±3.56 | 3.65±4.45 |
| 18. | Ethion | 0.3 ± 4.6 | 1.61±4.02 | 24.64±4.21 | 0.31±3.83 | 0.88±3.37 | 17.99±4.20 |
| 19. | Chlorpyrifos | 0.3 ± 4.7 | 5.84±10.02 | 12.13±2.37 | 0.44±4.87 | 3.37±4.51 | 12.89±2.54 |
| 20. | Carbosulfan | 0.4 ± 4.1 | 2.48±4.78 | 4.89±5.65 | 0.22±2.27 | 1.05±3.35 | 0.13±4.74 |
| 21. | Quinalphos | 0.5 ± 4.5 | 1.53±3.10 | 15.12±2.62 | 0.41±3.39 | 1.14±2.29 | 14.86±2.79 |
| 22. | Phosalone | 0.3 ± 4.9 | 2.87±4.59 | 12.02±2.50 | 0.28±3.63 | 1.95±3.01 | 21.73±4.20 |
| 23. | Chlormefos | 0.3 ± 3.6 | 1.94±3.40 | 8.54±6.25 | 0.36±3.64 | 1.62±2.48 | 24.13±4.75 |
| 24. | Diazinon | 1.3 ± 4.1 | 2.28±3.28 | 26.97±4.96 | 0.38±4.55 | 1.93±2.43 | 15.94±2.89 |
| 25. | Profenofos | 0.8 ± 3.8 | 2.96±6.00 | 23.20±4.61 | 0.48±4.41 | 1.06±2.51 | 21.71±4.32 |
| 26. | Acephate | 1.68±2.58 | 11.25±4.95 | 9.76±3.20 | 0.38±3.03 | 1.80±3.14 | 5.45±8.65 |
| 27. | Fenobucarb | 0.3 ± 4.7 | 1.49±4.79 | 0.91±4.14 | 0.19±2.33 | 0.64±3.23 | 0.66±4.17 |
| 28. | Iprovalicarb | 1.9 ± 3.4 | 0.07±4.96 | 12.81±4.94 | 1.63±2.88 | 0.04±4.89 | 9.38±4.87 |
| 29. | Methomyl | 4.6 ± 4.9 | 22.34±4.89 | 27.05±4.33 | 3.84±4.02 | 15.14±4.84 | 19.84±4.25 |
| 30. | Chlorfenvinphos | 8.1 ± 4.0 | 16.60±4.40 | 99.23±4.12 | 7.30±3.58 | 9.23±4.31 | 76.64±4.27 |

a) The values in the parentheses represent %RSD valves (n = 6), b) SD–Standard Deviation

spraying and serum 8-OHdG levels among the farmers diagnosed with three types of cancers as the regression coefficient $r^2$ value were observed to be 0.037, 0.020 and 0.006. Similarly, age and gender also did not show any significant association with the serum 8-OHdG levels among all the subjects studied in the present study (data not presented). However, Paired Student's t-test showed significant difference between the levels of serum 8-OHdG and dimethoate, quinalphos and pirimiphos residues in plasma samples of farmers diagnosed with leukemia and lymphoma cancers, while no such significant difference was observe among farm women diagnosed with breast cancer (Table 7).

It was interesting to note that the results of the parametric (Pearson) correlation analysis between the pesticide residues concentration and serum 8-OHdG levels among the farmers diagnosed with all three types of cancers (n = 180) showed significant difference only with the profenofos residues detected in plasma (Table 8).

**Table 5. Pesticides detected in the plasma samples of farmers, non-farmers and healthy controls (n = 360).**

| Cancer | Pesticide residues | Mean (ng/mL) | Range (ng/mL) |
|---|---|---|---|
| Leukaemia | Acephate + Dimethoate + Imidacloprid + Diazinon + Quinalphos + Malathion + Chlorfenvinfos (n = 1) | 148.81 | 1.38–800.49 |
| | Acephate + Dimethoate + Diazinon + Quinalphos (n = 1) | 26.12 | 14.93–22.40 |
| | Diazinon + Malathion + Chlorpyrifos (n = 1) | 366.51 | 60.19–764.53 |
| | Acephate + Diemethoate + Quinalphos +Chlorpyrifos (n = 1) | 25.14 | 5.07–58.99 |
| | Propoxur + Phenthoate + Ethion + Anilophos + Anilophos + Chlorfenvinfos (n = 1) | 858.32 | 1.31–1915.34 |
| | Dimethoate + Imidacloprid + Profenofos + Anilophos (n = 1) | 2.22 | 0.46–5.68 |
| | Dimethoate + Imidacloprid + Profenofos (n = 1) | 0.14 | 0.1–0.28 |
| | Carbaryl + Quinalphos (n = 1) | 20.72 | 0.28–4.17 |
| | Acephate + Quinalphos + Pirimiphos (n = 1) | 33.56 | 1.01–97.12 |
| | Acephate + Pirimiphos (n = 1) | 13.31 | 4.89–21.75 |
| Lymphoma | Pirimiphos + Quinalphos (n = 4) | 9.87 | 0.39–23.85 |

## Discussion

The present study results found that the farmers diagnosed with leukemia had higher levels of pesticide residues in their plasma than those diagnosed with lymphoma and breast cancers (farm women). It was interesting to note that, in addition to the residues of the approved pesticides such as chlorpyrifos, dimethoate, malathion, phosalone, and quinalphos, residues of some of the banned pesticides like monocrotophos, diazinon, and dichlorvos were also detected. There is some evidence to suggest that the exposure to certain pesticides may be associated with an increased risk of lymphoma, leukemia, and breast cancers [15, 16].

It was reported that extended duration of occupational exposure and long working hours in the pesticide treated fields may increase the health risks among the farmers [17]. In the present study, farmers who were classified into two groups based on their mean duration of exposure

**Table 6. Duration of exposure and mean levels of pesticide residues detected in plasma.**

| Pesticides | Leukemia | | Lymphoma | | Breast Cancer | |
|---|---|---|---|---|---|---|
| | Less than 14 years (Low) (ng/mL) | More than 14 years (High) (ng/mL) | Less than 20 years (Low) (ng/mL) | More than 20 years (High) (ng/mL) | Less than 18 years (Low) (ng/mL) | More than 18 years (High) (ng/mL) |
| *Acephate* | 35.75 | 21.75 | | | | |
| *Carbaryl* | 37.28 | - | | | | |
| *Propoxur* | 259.69 | - | | | | |
| *Dimethoate* | 5.24 | 0.49 | | | | |
| *Imidacloprid* | 48.09 | - | | | | |
| *Diazinon* | 90.98 | - | | | 17.64 | 2.11 |
| *Quinalphos* | 6.65 | 2.29 | | | | |
| *Pirimiphos* | 2.57 | 5.62 | 40.08 | 9.59 | | |
| *Phenthoate* | 597.01 | - | | | | |
| *Malathion* | 800.49 | 382.35 | | | | |
| *Chlorpyrifos* | 58.99 | 46.42 | | | | |
| *Profenofos* | 0.27 | 3.14 | | | | |
| *Ethion* | 34.17 | - | | | | |
| *Anilophos* | 1.12 | - | | | | |
| *Chrlorfenvinfos* | 981.25 | - | | | | |
| *Fenobucarb* | | | | | 17.2 | 23.6 |

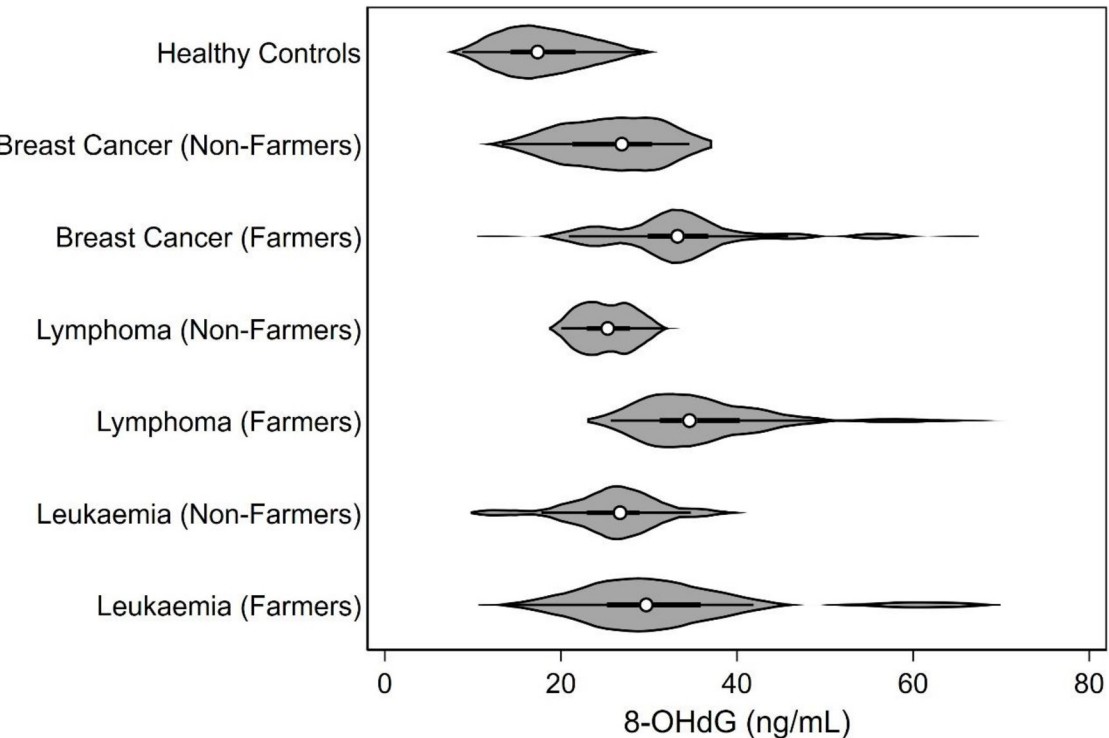

**Fig 3.** Levels of serum 8-OHdG in (1) farmers; (2) non-farmers and (3) control subjects.

to pesticides (i.e., high exposure and low exposure) showed a declining trend in the levels of pesticide residues in plasma in relation to the higher years of exposure. Further, it was found that the farmers above the age of 30 years were undergoing treatment for the past six months as compared to those below 30 years of age who were undergoing treatment for past15-30 days. Considering the fact that the duration of exposure in years is relative to the age of the participant, the phenomenon of lower levels of residues in their plasma samples with higher duration of exposure could be due to the fact that these subjects might be undergoing treatment for a longer time and might have not been engaged in spraying activities anymore immediately after they are diagnosed with three types of cancers studied as compared to the younger subjects with less duration of exposure who had shorter time span between diagnosis and the suspension of spraying activities. However, these findings were in contrast with the earlier studies [18, 19].

**Table 7. Pesticide residues in plasma and serum 8-OHdG levels among farmers.**

| Pesticide residues detected in the plasma | Leukemia (n = 60) | | | | Lymphoma (n = 60) | | | |
|---|---|---|---|---|---|---|---|---|
| | *n* | *Range ng/mL* | *8-OHdG ng/mL* | *p-value* | *n* | *Range ng/mL* | *8-OHdG ng/mL* | *p-value* |
| *Acephate* | 5 | 8.01–97.12 | 29.36±7.61 | 0.892 | - | - | - | - |
| *Dimethoate* | 6 | 0.28–16.13 | 31.69±5.61 | **0.028**\*\* | - | - | - | - |
| *Quinalphos* | 9 | 0.50–14.93 | 32.10±7.18 | **0.008**\*\* | 5 | 0.64–10.43 | 36.05±9.08 | **0.002**\*\* |
| *Pirimiphos* | 6 | 2.57±11.84 | 29.47±6.40 | 0.028 | 6 | 0.39–72.55 | 34.54±9.45 | **0.003**\*\* |

\*\*indicates significant difference $p<0.001$ at 95% CI

**Table 8. Pearson correlation between pesticide residues in plasma and serum 8-OHdG.**

| Quinalphos Pearson (p-value) | Fenobucarb Pearson (p-value) | Fenitrothion Pearson (p-value) | Malathion Pearson (p-value) | Profenofos Pearson (p-value) | Chlorpyrifos Pearson (p-value) | Ethion Pearson (p-value) |
|---|---|---|---|---|---|---|
| -0.044 (0.561) | -0.079 (0.292) | -0.11 (0.882) | 0.002 (0.975) | **0.197 (0.008)**** | 0.082 (0.277) | 0.139 (0.063) |

The presence of 8-OHdG, a biomarker for oxidative DNA damage, has been linked to exposure to certain pesticides suggesting that pesticides can cause DNA damage and increase the risk of cancer [20]. To understand the link between pesticide exposure and 8-OHdG levels, it is crucial to measure the actual levels of pesticide residues in biological samples. Pesticide residues in the blood reflect the body's exposure and the burden of the measured compounds [21]. In the current study, farmers had higher levels of serum 8-OHdG as compared to non-farmers and healthy controls, which are in consistence with previous studies [22, 23]. Interestingly, the present study found no significant association between serum 8-OHdG levels and confounding factors like age, duration of exposure, smoking, and consumption of tobacco and alcohol. These findings are in contrast with the previous studies [24–27]. However, the findings from the present study were in consistence by not finding any association with gender and 8-OHdG levels [28].

It was interesting to note that in the present study, a significant difference was observed between the profenofos residues in plasma and serum 8-OHdG levels which are similar to that reported earlier showing profenofos intoxication leading to oxidative stress in experimental animals [29]. It is a well-known fact that oxidative stress is a contributory factor for carcinogenesis [13]. The present study results highlight the importance of the interesting observations made on the potential negative impacts of profenofos exposure and its ability to induce oxidative stress and the need for further in-depth investigation.

Previous studies have suggested that organophosphorus pesticides can cause oxidative stress by producing free radicals, lipid peroxidation affecting antioxidant or ROS scavenging enzymes [30]. The present study also found higher levels of serum 8-OHdG in majority of farmers who reported using organophosphorus pesticides. These findings are to some extent consistent with a study conducted in Lucknow, Uttar Pradesh, India with elevated levels of urinary 8-OHdG among farmers using organophosphate pesticides [31]. Further, the use of personal protective equipment (PPE) was found to have reduced the levels of 8-OHdG in a study conducted elsewhere [32]. However, none of the farmers in the present study reported to have used PPE while spraying pesticides. The present study results were consistent with earlier studies, highlighting the need for appropriate safety measures and educational interventions [33].

Of late, the use of 8-OHdG in molecular epidemiology studies as a biomarker for carcinogenesis is increasing. The findings from the present study suggests that farmers diagnosed with leukemia, lymphoma and breast cancers and were occupationally exposed to pesticides prior to their diagnosis with cancer showed increased oxidative stress levels in the form of serum 8-OHdG when compared to the non-farmers/unexposed diagnosed with similar types of cancers and healthy control groups who neither had any history of exposure to pesticides nor diagnosed with any cancer. Although the presence of prevailing carcinoma cells and other contributory factors such as age, status of smoking, tobacco and alcohol consumption, exposure to other environmental chemicals and pollutants could also play a role in increasing the serum 8-OHdG levels, the possibility of elevated serum 8-OHdG levels due to the exposure to pesticides cannot be ruled out.

Accumulated pesticide residues in biological matrices are hypothesized as a possible risk factor for carcinogenesis. Previously, several attempts were made to shed light on the role of pesticides in the causation of various cancers, however, the results have been largely

ambiguous [34]. However, it's worth noting that the risk of developing cancer is influenced by a wide range of factors, including genetics, lifestyle choices, and environmental exposures, to various other pollutants hence cannot be a specific cause. Additionally, it's important to note that the evidence linking pesticide exposure to these cancers is not conclusive, and more research is needed to fully understand the underlying mechanisms of carcinogenesis. The present study is only an attempt to demonstrate that pesticides residues were detected in the plasma samples of the farmers diagnosed with all three types of cancers which is of significant importance from the public health perspective and emphasize the need for essential pre-requisites such as adoption of safety measures viz., PPE and intervention programs to enhance the knowledge, attitude and practices of farmers engaged in pesticide spraying activities. In the present study, the farmers showed higher levels of serum 8-OHdG as compared to the non-farmers and healthy controls along with the significant difference between detected pesticide residues and serum 8-OHdG levels. The present study findings may facilitate in developing strategies to decrease the oxidative damage due to exposure to pesticides by using of dietary antioxidants supplements [35]. Moreover, the present study also made an attempt to address the importance of the detection of not only the approved pesticide residues but also those that have been banned for usage, thereby suggesting the need for cohort and follow-up studies involving different types of cancers and with more in-depth molecular parameters. In conclusion, the study highlights the detection of both approved and banned pesticide residues among all the farmers diagnosed with lymphoma, leukaemia and breast cancers while no residues were detected among the non-farmers and healthy controls. However, a significant association between the profenofos residues and serum 8-OHdG was also observed in the present investigation.

## Supporting information

**S1 File.**
(XLSX)

## Acknowledgments

The authors acknowledge the support provided by Dr. R. Hemalatha, Director, ICMR-NIN and Dr. N. Jayalatha, Director, MNJ Institute of Oncology & Regional Cancer Institute, Hyderabad and the healthcare officials and paramedical staff, for their helpful co-operation in conducting the study.

## Author Contributions

**Conceptualization:** Padmaja R. Jonnalagadda.

**Data curation:** Summaiya Lari, Balakrishnan Senthil Kumar.

**Formal analysis:** Arun Pandiyan, Janardhan Vanka, Balakrishnan Senthil Kumar.

**Investigation:** Arun Pandiyan.

**Methodology:** Janardhan Vanka, Sudip Ghosh.

**Project administration:** Padmaja R. Jonnalagadda.

**Resources:** Babban Jee.

**Supervision:** Padmaja R. Jonnalagadda.

**Validation:** Summaiya Lari, Babban Jee.

**Writing – original draft:** Arun Pandiyan.

**Writing – review & editing:** Summaiya Lari.

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
