## [Decision Letter · Decision Letter 0]

21 Aug 2023

PONE-D-23-18182Plasma pesticide residues – serum 8-OHdG among farmers/non-farmers diagnosed with lymphoma, leukaemia and breast cancers: a case-control studyPLOS ONE

Dear Dr. Jonnalagadda,

Thank you for submitting your manuscript to PLOS ONE. After careful consideration, we feel that it has merit but does not fully meet PLOS ONE’s publication criteria as it currently stands. Therefore, we invite you to submit a revised version of the manuscript that addresses the points raised during the review process.

We look forward to receiving your revised manuscript.

Kind regards,

Naji Arafat Mahat, PhD

Academic Editor

PLOS ONE

“This work was supported

by Department of Health Research, Ministry

of Health & Family Welfare, Government

of India under grant R.11012/17/2017-HR”

“The authors would like to thank the Department of Health Research, Ministry of Health & Family Welfare, GoI, for providing the financial assistance. The authors also acknowledge the support provided by Dr. R. Hemalatha, Director, ICMR-NIN and Dr. N. Jayalatha, Director, MNJ Institute of Oncology & Regional Cancer Institute, Hyderabad and the healthcare officials and paramedical staff, for their helpful co-operation in conducting the study.”

“This work was supported

by Department of Health Research, Ministry

of Health & Family Welfare, Government

of India under grant R.11012/17/2017-HR”

6. Please include a separate caption for each figure in your manuscript.

7. Please upload a copy of Figure 3.1, to which you refer in your text on page 11. If the figure is no longer to be included as part of the submission please remove all reference to it within the text.

Reviewers' comments:

Reviewer's Responses to Questions

**Comments to the Author**

1. Is the manuscript technically sound, and do the data support the conclusions?

Reviewer #1: Partly

Reviewer #2: Yes

2. Has the statistical analysis been performed appropriately and rigorously? 

Reviewer #1: Yes

Reviewer #2: Yes

3. Have the authors made all data underlying the findings in their manuscript fully available?

Reviewer #1: Yes

Reviewer #2: Yes

4. Is the manuscript presented in an intelligible fashion and written in standard English?

Reviewer #1: Yes

Reviewer #2: Yes

5. Review Comments to the Author

Reviewer #1: The work is very interesting. However, the authors did not describe the extraction and clean up procedures for the samples properly. The validation was not done correctly. The article has notable merits. Therefore, I think that the article is suitable for publication after necessary clarifications and modifications.

Specific Comments:

In page 7, line 27: Please write ….The pesticide residues detected were chlorpyrifos, dimethoate…..

In page 11, line 144-149: The extraction and cleanup procedures did not written properly. Please write the extraction procedures clearly. The amount of sample that the authors have taken for extraction and cleanup, and for liquid liquid extraction how much ACN was taken? It was correct that 200 uL ACN taken for liquid liquid extraction? For reconstraction why the authors have used only 200 uL ACN, it was too low?

In page 13, line 167-175: The procedures for LOQ estimation was not appropriate. According to SANTE and CODEX guidelines, the LOQ should be the lowest fortification level that is achieved the acceptable accuracy and precission. In the study, the authors did not follow it. In this study, the lowest forfication level was 10 ng/mL. Therefore, the LOQ should be 10 ng/mL. If the authors claim that the LOQ was 2 ng/mL , why they did not spike this amount? It should be considered.

It would better for the readers if the authors provide any justification why they have found higher levels of pesticide residues in the farmers who have leukemia? Is there any relationship among the nature of cancer?

In table 1.3: The LOQ was 2-10 ng/mL. However, in table 1.5, the authors have presented posiive samples containing residues below the LOQ levels (0.28- 1.38 ng/mL) . It was not appropriate. The positive samples should have residues at least LOQ levels or above LOQ levels. The Positive samples are not estimated based on the LOD. Therefore, it is necessary to change it.

In table 1.3: The linear range was 1-2000 ng/mL, it means that the LCL was 1 ng/mL. How the authors have quantify below that levels (0.28- 1.38 ng/mL; in table 1.5)?

Reviewer #2: 1. Should include 8-OHdG in abstract and intro. Why did the author use 8-OHdG as markers of damage instead of other markers? Lack of explanation about this.

2. Abstract: The pesticide residues in farmers vs non-farmers, significant or not?

3. Please include ethical approval number/reference.

4. Research methodology: What is the percentage/number of non-farmers with similar cancers as the study population ie. the farmers?

5. What are the occupation of non-farmers and control populations?

6. Should explain what is pucca, semi-pucca and kutcha for the benefit of readers.

7. the location away from farm, what is the operational definition?

6. PLOS authors have the option to publish the peer review history of their article (what does this mean?). If published, this will include your full peer review and any attached files.

Reviewer #1: **Yes: **Dr. Mohammad Dalower Hossain Prodhan

Reviewer #2: No

---

## [Author Response · Author response to Decision Letter 0]

13 Nov 2023

Editor comments: 

Comment 1: Please ensure that your manuscript meets PLOS ONE's style requirements, including those for file naming. 

Yes, the revised manuscript meets PLOS ONE's style requirements, including those for file naming.

Comment 2: Thank you for stating the following financial disclosure:

“This work was supported by Department of Health Research, Ministry of Health & Family Welfare, Government of India under grant R.11012/17/2017-HR”

Authors wish to mention that the funding agency had no role except that it provided financial grant to undertake the study. Hence the following statement has been included in the cover letter:

Comment 3: Thank you for stating the following in the Acknowledgments Section of your manuscript:

“The authors would like to thank the Department of Health Research, Ministry of Health & Family Welfare, GoI, for providing the financial assistance. The authors also acknowledge the support provided by Dr. R. Hemalatha, Director, ICMR-NIN and Dr. N. Jayalatha, Director, MNJ Institute of Oncology & Regional Cancer Institute, Hyderabad and the healthcare officials and paramedical staff, for their helpful co-operation in conducting the study.”

“This work was supported by Department of Health Research, Ministry of Health & Family Welfare, Government of India under grant R.11012/17/2017-HR” Please include your amended statements within your cover letter; we will change the online submission form on your behalf.

As per the Editors’ suggestion, the funding-related text has been removed from the Acknowledgments section or other areas of the manuscript (Page 25; Line number 434; Section Acknowledgments). However, as per the journal format, this could appear in the funding statement section of the online submission form.

The Funding Statement stands correct. No update is required in the same.

Comment 4: In your Data Availability statement, you have not specified where the minimal data set underlying the results described in your manuscript can be found. PLOS defines a study's minimal data set as the underlying data used to reach the conclusions drawn in the manuscript and any additional data required to replicate the reported study findings in their entirety. All PLOS journals require that the minimal data set be made fully available. For more information about our data policy, please see http://journals.plos.org/plosone/s/data-availability.

The supporting information file is uploaded along with manuscript as a supplementary file. However, as per the ICMR guidelines, no patient’s background information can be provided as the study was undertaken after obtaining institutional ethical clearance from both ICMR-NIN and MNJ (a referral State Government) hospital. 

Comment 5: PLOS requires an ORCID iD for the corresponding author in Editorial Manager on papers submitted after December 6th, 2016. Please ensure that you have an ORCID iD and that it is validated in Editorial Manager. To do this, go to ‘Update my Information’ (in the upper left-hand corner of the main menu), and click on the Fetch/Validate link next to the ORCID field. This will take you to the ORCID site and allow you to create a new iD or authenticate a pre-existing iD in Editorial Manager. Please see the following video for instructions on linking an ORCID iD to your Editorial Manager account: https://www.youtube.com/watch?v=_xcclfuvtxQ

As suggested, the ORCID ID of the corresponding author has been updated in the Editorial Manager.

Comment 6: Please include a separate caption for each figure in your manuscript.

As per the Editors’ suggestion, a separate caption for each figure was added in the revised manuscript (Page 27; Line numbers 448-450; Section Figure Captions).

Comment 7: Please upload a copy of Figure 3.1, to which you refer in your text on page 11. If the figure is no longer to be included as parts of the submission please remove all reference to it within the text.

We regret the inadvertent error. The comment is now addressed and the figure is removed and all the figure captions are corrected. 

Reviewer #1 Comments: 

Comment 1: In page 7, line 27: Please write….The pesticide residues detected were chlorpyrifos, dimethoate…..

As suggested, the corrections were made in the revised manuscript (Page 1; Line numbers 25 & 26).

Comment 2: In page 11, line 144-149: The extraction and cleanup procedures did not written properly. Please write the extraction procedures clearly. The amount of sample that the authors have taken for extraction and cleanup, and for liquid liquid extraction how much ACN was taken? It was correct that 200 uL ACN taken for liquid liquid extraction? For reconstraction why the authors have used only 200 uL ACN, it was too low?

Yes, it is correct that 200 µl of acetonitrile was taken for liquid-liquid extraction; acetonitrile was added for protein precipitation. The protein precipitation is generally carried out with 1:1 ratio of plasma : organic solvent used. Here, we have also followed a similar methodology. After protein precipitation, the samples were centrifuged to remove the protein pellet and the supernatant extract is completely evaporated to dryness. The residue was reconstituted in 200 µL acetonitrile. As the extraction work was carried out in the microcentrifuge vials, it is possible to reconstitute the residue using 200 µL of acetonitrile, and vortexing followed by centrifugation and also to facilitate the smooth extraction and subsequent transfer of extract which is more than sufficient for LC-MS analysis as we need only 10 µL of sample for injection. The reconstitution value was kept equal to the initial volume of the plasma, so that the additional corrections for concentrations can be avoided. The obtained values directly represent the concentration of pesticide residues present in the plasma samples. As suggested by the reviewer, extraction and cleanup procedures have been properly re-written in the revised manuscript (Pages 5 & 6; Line numbers 146-176).

Comment 3: In page 13, line 167-175: The procedures for LOQ estimation was not appropriate. According to SANTE and CODEX guidelines, the LOQ should be the lowest fortification level that is achieved the acceptable accuracy and precision. In the study, the authors did not follow it. In this study, the lowest forfication level was 10 ng/mL. Therefore, the LOQ should be 10 ng/mL. If the authors claim that the LOQ was 2 ng/mL, why they did not spike this amount? It should be considered.

Authors agree with the reviewer’s suggestion that according to SANTE and CODEX guidelines, the LOQ should be the lowest fortification level that is achieved the acceptable accuracy and precision. However, in the present study, the LOQ values were calculated correctly in this method. This information was earlier provided in the manuscript in line number 166 and 167 of page 6, where the concentrations used for linear calibration curve were clearly mentioned. The linearity was conducted from 1 to 2000 ng/mL concentration levels. The LOQ values were identified where the precision for 5 numbers of samples falls within 20% deviation levels (as per SANTE, CODEX, FDA and ICH Q2R1 guidelines). Hence in the table, the values were mentioned as 2-10 ng/mL. Therefore, the given values are justified.

Comment 4: It would better for the readers if the authors provide any justification why they have found higher levels of pesticide residues in the farmers who have leukemia? Is there any relationship among the nature of cancer?

Interestingly, the subjects diagnosed with leukemia had higher levels of pesticide residues. However, a clear-cut relationship with leukemia could not be established possibly these subjects may have slow/low clearance rate of these pesticide residues as compared to other types of cancers. 

Comment 5: In table 1.3: The LOQ was 2-10 ng/mL. However, in table 1.5, the authors have presented positive samples containing residues below the LOQ levels (0.28- 1.38 ng/mL). It was not appropriate. The positive samples should have residues at least LOQ levels or above LOQ levels. The Positive samples are not estimated based on the LOD. Therefore, it is necessary to change it.

The authors would like to clarify the reviewer that in the present study the concentration values in the plasma samples were calculated with regression equation obtained in the linear calibration curve. Hence, the values were obtained below LOQ levels. However, for confirmation, the studies were also conducted using 600 µL of plasma samples and protein precipitation was carried out with 600 µL of acetonitrile and the sample was centrifuged at 10000 rpm for 5 min, the supernatant solution was evaporated to dryness and the residue was reconstituted using 200 µL of acetonitrile. This is 3 times concentration and the values will fall within the linear calibration curve levels. The instrument linearity is 1-2000 ng/mL, however, for confirmation with respect to clinical samples, if the concentration of analytes was not within the detectable range, 600 µL of plasma samples (3x concentration) was used. The same is now incorporated in the revised manuscript (line no. 173 – 176). 

Comment 6: In table 1.3: The linear range was 1-2000 ng/mL, it means that the LCL was 1 ng/mL. How the authors have quantify below that levels (0.28- 1.38 ng/mL; in table 1.5)?

Authors agree with the reviewer’s observation that the LCL for the method was 1 ng/mL. However, we would like to mention that the lowest concentration can be analysed with the same method by increasing the initial volume of the plasma samples. Hence, the presented levels are justified.

Reviewer #2 Comments: 

Comment 1: Should include 8-OHdG in abstract and intro. Why did the author use 8-OHdG as markers of damage instead of other markers? Lack of explanation about this.

A brief introduction and an explanation on the rationale for studying 8-OHdG in the present study are included in the revised manuscript along with relevant references as suggested (Page 3; Line numbers 71-77).

Comment 2: Abstract: The pesticide residues in farmers vs non-farmers, significant or not?

Authors wish to mention that in the present study, none of the non-farmers and the healthy controls were detected with any pesticide residues in the plasma samples and hence the reason no statistical comparison could not be performed. The same is reflected in the present manuscript in the section abstract (Page 1; Line numbers 29 & 30).

Comment 3: Please include ethical approval number/reference.

As suggested, the ethical approval number/reference numbers are included in the revised manuscript (Page 3; Line numbers 85-87).

Comment 4: Research methodology: What is the percentage/number of non-farmers with similar cancers as the study population i.e. the farmers?

The number of non-farmers with similar cancers as the study population (farmers) were 90 [non-farmers diagnosed with leukemia (n=30), lymphoma (n=30) and breast cancer (n=30)]. The exact number of farmers and non-farmers diagnosed with each cancer are now included in the revised manuscript (Page 4; Line numbers 105-111).

Comment 5: What are the occupation of non-farmers and control populations?

The occupation of the non-farmers and control population ranged across various sectors such as transport, household services, etc. except getting engaged in any of the farming activities. We would like to emphasize that careful consideration was given while collecting the questionnaire data and samples that none of the subjects who had history of exposure to pesticides were included under the category of non-farmers and control groups. However, the healthy controls are subjects belonging to the family members of farmers but with no exposure to pesticides at any given point of time (Page 4; Line numbers 111-114).

Comment 6: Should explain what is pucca, semi-pucca and kutcha for the benefit of readers.

Inadvertent error is highly regretted. The parameters for ‘pucca’, ‘semi-pucca’ and ‘kutcha’ are explained as foot notes in the Table 1.2 (Page 12; Line numbers 275-276).

Comment 7: The location away from farm, what is the operational definition?

Since establishing such operational definitions would make the data collection process tedious as it takes time to explain the farmers each question, no such operational definitions were followed in the present study. Hence, the location of the house was considered “In the farm”, if the house was located within the field or at close proximity to the field, the exposure to pesticides would be relatively more as compared to those living “Away from the farm”.

Reviewer #3 Comments: 

Comment 1: While the chosen topic is captivating, the paper occasionally confuses the study design between cross-sectional and case-control methods. A case-control study should be presented Odds Ratio and 95% confidence interval (95% CI) associated with the exposure covariates by fitting multiple logistic regression models. Logistic regression modeling, in its various forms, has become by far the most frequently applied method for multivariable analysis of case-control studies. Additionally, the inclusion of matching factors during sample recruitment is essential.

We highly appreciate the recommendations. The standard study design of “Hospital based case control study” is followed throughout the revised manuscript. As per the suggestion, we performed both multiple regression and logistic regression and we did not find any significant association between pesticide residue levels or serum 8-OHdG and other confounding variables such as duration of exposure, age, gender, usage of PPE, dietary habits, personal habits such as smoking, drinking alcohol, consumption of tobacco etc. 

Comment 2: In the Methodology section, critical details are missing. To bolster transparency, it’s crucial to include the ethical approval number (line 81).

As suggested, the ethical approval numbers are included in the revised manuscript (Page 3; Line numbers 85-87).

Comment 3: Further clarification is required regarding how samples were recruited from non-farmers and healthy individuals. Detailed 

---

## [Decision Letter · Decision Letter 1]

23 Nov 2023

Plasma pesticide residues – serum 8-OHdG among farmers/non-farmers diagnosed with lymphoma, leukaemia and breast cancers: a case-control study

PONE-D-23-18182R1

Dear Dr. Jonnalagadda,

We’re pleased to inform you that your manuscript has been judged scientifically suitable for publication and will be formally accepted for publication once it meets all outstanding technical requirements.

Kind regards,

Naji Arafat Mahat, PhD

Academic Editor

PLOS ONE

Additional Editor Comments (optional):

Reviewers' comments:

Reviewer's Responses to Questions

**Comments to the Author**

1. If the authors have adequately addressed your comments raised in a previous round of review and you feel that this manuscript is now acceptable for publication, you may indicate that here to bypass the “Comments to the Author” section, enter your conflict of interest statement in the “Confidential to Editor” section, and submit your "Accept" recommendation.

Reviewer #1: (No Response)

2. Is the manuscript technically sound, and do the data support the conclusions?

Reviewer #1: (No Response)

3. Has the statistical analysis been performed appropriately and rigorously? 

Reviewer #1: (No Response)

4. Have the authors made all data underlying the findings in their manuscript fully available?

Reviewer #1: (No Response)

5. Is the manuscript presented in an intelligible fashion and written in standard English?

Reviewer #1: (No Response)

6. Review Comments to the Author

Reviewer #1: The authors have revised the manuscript following the reviewer guidelines. Therefore, the article can be accepted for publication

7. PLOS authors have the option to publish the peer review history of their article (what does this mean?). If published, this will include your full peer review and any attached files.

Reviewer #1: **Yes: **Dr. Mohammad Dalower Hossain Prodhan

---

## [Editor Report · Acceptance letter]

5 Aug 2024

PONE-D-23-18182R1 

PLOS ONE

Dear Dr. Jonnalagadda, 

I'm pleased to inform you that your manuscript has been deemed suitable for publication in PLOS ONE. Congratulations! Your manuscript is now being handed over to our production team.

Kind regards, 

on behalf of

Dr. Naji Arafat Mahat 

Academic Editor

PLOS ONE